# Strategies for Assessing Arbovirus Genetic Variability in Vectors and/or Mammals

**DOI:** 10.3390/pathogens9110915

**Published:** 2020-11-05

**Authors:** Camille Victoire Migné, Sara Moutailler, Houssam Attoui

**Affiliations:** 1UMR BIPAR, Laboratoire de Santé Animale, ANSES, INRAE, Ecole Nationale Vétérinaire d’Alfort, Paris-Est Sup, 94700 Maisons-Alfort, France; camille.migne.ext@anses.fr; 2UMR1161 Virologie, INRAE, ANSES, Ecole Nationale Vétérinaire d’Alfort, Paris-Est Sup, 94700 Maisons-Alfort, France

**Keywords:** arboviruses, arthropods, genetic variability, insect-borne virus, tick-borne virus

## Abstract

Animal arboviruses replicate in their invertebrate vectors and vertebrate hosts. They use several strategies to ensure replication/transmission. Their high mutation rates and propensity to generate recombinants and/or genome segment reassortments help them adapt to new hosts/emerge in new geographical areas. Studying arbovirus genetic variability has been used to identify indicators which predict their potential to adapt to new hosts and/or emergence and in particular quasi-species. Multiple studies conducted with insect-borne viruses laid the foundations for the “trade-off” hypothesis (alternation of host transmission cycle constrains arbovirus evolution). It was extrapolated to tick-borne viruses, where too few studies have been conducted, even though humans faced emergence of numerous tick-borne virus during the last decades. There is a paucity of information regarding genetic variability of these viruses. In addition, insects and ticks do not have similar lifecycles/lifestyles. Indeed, tick-borne viruses are longer associated with their vectors due to tick lifespan. The objectives of this review are: (i) to describe the state of the art for all strategies developed to study genetic variability of insect-borne viruses both in vitro and in vivo and potential applications to tick-borne viruses; and (ii) to highlight the specificities of arboviruses and vectors as a complex and diverse system.

## 1. Introduction

### 1.1. Vector-Borne Diseases in the World

More than half of the world is at risk from vector-borne diseases. They are illnesses caused by pathogens in human and animal populations. Each year more than one billion humans are infected and more than one million die from vector-borne diseases. The most affected by these diseases are the poorest and least-developed countries [1]. Hematophagous arthropods (mosquitoes, ticks, sandflies, triatomine bugs, fleas and flies) transmit a wide variety of pathogens including bacteria, parasites and viruses [1]. In addition to the public health burden, vector-borne diseases negatively impact the economies of affected countries. For example, according to the WHO, the annual economic costs of malaria in Africa have been estimated to be about $12 billion [2]. In animal health, vector-borne diseases cause important economic losses particularly in the breeding industry. For instance, the *Culicoides-*borne bluetongue virus (BTV) is responsible for mortality and morbidity in sheep. The economic impact of BTV outbreaks can be substantial. In 2006, a BTV outbreak in Europe cost $1.4 billion and $85 million to France and the Netherlands, respectively [3].

Pathogen emergence is influenced by a range of parameters including pathogen evolution, the role of arthropod vectors and their feeding preferences (anthropophilic and/or ornithopophilic) in global or local geographic expansion, anthropogenic factors (human movements, etc.) and climate change [4,5]. In this review, we emphasise the role of ticks as arthropod vectors, particularly for viruses. Studying the genetic variability of tick-borne viruses falls far behind mosquito-borne viruses. Studies conducted with insect-borne viruses have set the methodology necessary for assessing genetic variability of tick-borne viruses.

### 1.2. Ticks and Tick-Borne Disease

Approximatively 900 tick species were identified and classified into three families: *Argasidae* (soft ticks, 183 species), *Ixodidae* (hard ticks, 683 species) and *Nuttalliellidae* (one species). They are obligate hematophagous ectoparasites and feed on vertebrate hosts [6]. Ticks are found all around the world and affect animal and human health. They are responsible for significant economic losses, especially in livestock [7]. Ticks are involved in vector-borne diseases. Indeed, ticks can transmit a wide variety of pathogens such as bacteria, parasites and viruses. They are considered as the primary vectors for pathogens in animal health, and, in terms of human health, they are second to mosquitoes [8]. When ticks feed on infected vertebrate hosts, they likely ingest microorganisms. During their life cycle, hard ticks are in stasis each time they take a blood meal. For example, the European hard tick *Ixodes ricinus* is in stasis three times during its life cycle: larval, nymphal and adult stages. For ticks to act as vectors, the pathogen must be trans-stadially transmitted, thus nymphs and adults are major vectors [9]. Vector competence is an essential property for a tick to be considered a vector. Following ingestion, a pathogen must replicate within the arthropod in order to be transmitted to a new host during the next blood meal. Finally, tick population size is an important criterion which should be taken into account when discussing vector capacity [9].

Less than 10% of known ticks are involved in field transmission of viruses. These ticks belong to genera *Ornithodoros* and *Argas* (soft ticks) within family *Argasidae* and genera *Ixodes, Haemaphysalis, Hyalomma, Amblyomma, Dermacentor, Rhipicephalus* and *Boophilus* (hard ticks) within family *Ixodidae*. Certain ticks can transmit several viral species. Due to their lifespan and the potential for transovarial transmission of certain tick-borne, ticks could play a role of reservoir for the viruses they transmit [10]. In total, 170 tick-borne viruses have been identified to date (Table 1), as described in several reviews [10,11].

Despite many tick-borne viruses are responsible for human infections, only a few have been studied, in particular tick-borne encephalitis (TBE) and Crimean-Congo haemorrhagic fever (CCHF) virus. Table 2 lists some tick-borne viruses that have been responsible for outbreaks in humans or animals [11,12,13]. 

## 2. Arboviruses

At least 500 arboviruses (arthropod-borne viruses) have been identified to date, including tick-borne viruses. Almost half of arboviruses are transmitted by mosquitoes, a third by ticks and the rest by sandflies and biting midges [10]. Arboviruses do replicate in two hosts: their invertebrate vector and vertebrate host. Vectors and vertebrate hosts can play a role of amplifier and/or reservoir [34]. These viruses have important impacts on human and animal health. They are either epizootic or zoonotic. Emergence and/or re-emergence of arboviruses is a challenge that humanity is facing during the current century [4]. Incursions into novel geographical regions are becoming more frequent. For example, bluetongue disease caused by BTV infection has been considered for a long time as a tropical disease based on its geographical distribution. It was initially described within in a geographical region between approximately 40° N and 35° S. During the second half of the 20th century, it emerged all around the world, disrupting trade and causing severe economic damage [35].

### 2.1. Viruses Quasi-Species

Almost all animal arboviruses are RNA viruses, except African swine fever virus (DNA virus), and they use several strategies to ensure their replication and transmission. Mutation rates of RNA viruses are 300-fold higher than for DNA viruses. Substitutions during replication have been estimated within the range of 10^−3^–10^−5^ substitutions per nucleotide per replication [36]. The genetic variation of RNA viruses has to be considered for three levels: single replicative unit (early events when mutants are generated), gathering of several units (recombination, gene duplication, genome segment reassortments, gene transfers, etc.) and infection of susceptible hosts. Generation times for RNA viruses are short and the size of generated populations (quasi-species) is large [37]. In a quasi-species, viral progeny from a single cell is heterogeneous and non-identical but they have closely related genomes [38].

Mutants can be compartmentalised in different organs of a single organism. There are two mathematical models which are pertinent to study and interpret behaviour of virus quasi-species: calculation of the concentration of copies with no changes and consequently mutants and formulations of the error threshold for maintenance of genetic information [37].

Viruses are subject to competitive selection and random events. Some deleterious mutations lead to the extinction of given variants as their fitness declines. Natural selection is far more efficient for eliminating mutations with larger rather than smaller impact. In addition, viruses are confronted with bottlenecks. Genetic bottlenecks are the events that can cause important reduction of a population size, reducing their genetic diversity. Low frequency genomes are eliminated when they are subjected to a bottleneck [37]. For example, a study of Venezuelan equine encephalitis virus (VEEV) showed that the midgut of *Culex taeniopus* mosquito was a severe bottleneck. The size of the viral population was significantly reduced after passage through the intestinal barrier [39]. The review by Forrester et al. describes the multiple bottlenecks to which West Nile virus (WNV), VEEV and CHIKV are subjected to upon infection of mosquito vectors [40]. Repeated bottlenecks lead to changes in the genomes of RNA viruses and the emergence of novel variants [41]. If a single genome from a viral quasi-species can generate a new viral population, this mutant probably has an advantageous mutation. This mutation is inheritable by the next generation. Consequently, bottlenecks are responsible for the accumulation of mutations in the consensus sequence of a quasi-species [42]. RNA viruses can cause acute and/or chronic infections. Viral quasi-species play a role in the escape from host immune response, therapeutic treatments and vaccines. Consequently, they play a role in the progression of pathogenesis and disease as shown for some viruses like DENV [43].

### 2.2. Genetic Variability Studies of Insect-Borne Viruses

RNA viruses can expand their host range and adapt to new environment. The study of genetic variability helps understanding and anticipating viral emergence. For example, in 2004, Chikungunya virus (CHIKV) caused major outbreaks in the Indian Ocean, India, Malaysia and Sri Lanka. The virus, which was responsible for these outbreaks, was identified as the African strain of CHIKV usually transmitted to humans by *Aedes aegypti* mosquitoes. The dispersal of this virus from Central/East Africa to Comoros Islands occurred via infected mosquitoes and/or humans. As a result of a single amino acid substitution (A226V) in the E1 envelope glycoprotein, CHIKV was able to infect *Aedes albopictus,* mosquitoes found in La Réunion Island [5]. This change in CHIKV has been described to lower the threshold of viraemia necessary to infect *Ae. albopictus* mosquitoes. A change at this position in Semliki Forest virus (P226S) was previously reported to free the virus from cholesterol dependence [44].

Several studies were conducted with insect-borne viruses in order to understand their genetic variability and fitness. Among those studied, there are viruses belonging to genera *Flavivirus* (WNV, Rabensburg virus (RBGV), *Saint Louis encephalitis virus* (SLEV), *Dengue virus* (DENV) and *Zika virus* (ZIKV)), *Alphavirus* (*Eastern equine encephalitis virus* (EEEV), VEEV, *Sindbis virus* (SINV), *Chinkungunya virus* (CHIKV) and *Ross river virus* (RRV)), *Phlebovirus* (*Rift valley fever virus* (RVFV)), *Rhabdovirus* (*Vesicular stomatitis virus* (VSV)) and *Orbivirus* (BTV). The “trade-off” hypothesis, which came as a consequence of these studies, proposes that the alternating host transmission cycle of arbovirus likely constrains their evolution. These studies provide the basis for assessing trade-off in tick-borne viruses and better understanding their emergence. This review focusses on results which were highlighted by previous studies. Both in vitro and in vivo approaches were developed in these studies.

#### 2.2.1. In Vitro Studies

Classically, in vitro studies are designed as shown in Figure 1 with serial and alternated passages in mammalian/avian and arthropod cell lines. Various tests are performed to assess fitness, virulence and genetic aspects. Some of these studies result in converging hypotheses. For example, during serial passages, viruses tend to become specialists when they are grown in a single-host cell type, whereas in alternated passages viruses have a similar behaviour as the parental strain. This characteristic was observed with different viruses: RVFV, EEEV, SINV, VEEV, SLEV and WNV [45,46,47,48,49]. Often, adaptation to a single cell type resulted in faster growth kinetics within the same cell type. This is the case of RVFV and SINV, where mammalian cell-adapted strains show faster replication rates as compared to the parental strain, whereas, in mosquito cells, viral titers are lower [45,47]. Following 40 passages in mosquito cells (C6/36), WNV and SLEV showed increased fitness and replication in this cell type [49]. For other viruses such as VEEV, the adaptation was demonstrated by an increase in binding efficiency to mammalian cells after serial passages in this cell type. Indeed, an amino acid substitution (G3K) was identified in the E2 glycoprotein. This position corresponds to the furin cleavage site which is involved in binding to heparan sulfate [48]. Deep sequencing and computational analyses of ZIKV envelope protein (E protein) were performed after transfecting cDNA libraries with codon substitutions into C6/36 or Vero cells. Specific substitutions conferred advantages to replicate in mosquito or primate cells, while other substitutions had negative impact on replication [50]. In a study focussing on RBGV by Ngo et al. (2019) [51], the authors successfully adapted the virus to a vertebrate host. RBGV was classified as lineage 3 of WNV and has never been isolated from vertebrate hosts. After four passages of two strains of RBGV in HEK-293 cells, at increasing temperatures (from 28 to 35 °C), they obtained variants capable of infecting and replicating in vertebrate cells. The majority of changes were found in two genes (NS3 and NS5 genes), which were presumed to play a role in host specificity and fitness. Using reverse genetics, they assessed the impact of mutating NS3 position 5716 (A5716G substitution, identified during the in vitro experimental infections). They tested the potential effect of this substitution on host range. A modest decrease of virus titers in HEK-293 cells was observed at both 28 and 35 °C [51].

Serial passages reduce the capacity of arboviruses to infect heterologous cell types, resulting in a decreased fitness. CHIKV resulting from seven continuous passages in BHK-21 cells had a greater fitness in BHK-21 than in C6/C36 cells as compared to the parental strain. In addition, the alternated strain (C6/36–BHK-21 and C6/36–HeLa) showed an increased fitness in both cell types as compared to the parental strain [52]. Fitness of VSV varies in a cell-type dependent manner. Indeed, during a persistent infection of sanfly cells (LL-5), the fitness declined as early as the first passage, and then slightly increased, reaching a level of stability after passage 20. By contrast, during serial passages in BHK-21 cells, fitness continuously increased. In alternated passages, fitness initially increased slightly, and then declined, reaching a level of relative stability around passage 20 [53]. Virulence of certain viruses is decreased when they are grown in a single host cell type. The gene encoding phosphoprotein *NSs*, which is responsible for RVFV virulence, is deleted after 30 serial passages in BHK-21 or Aag2 cells, while, during alternated passages, it is not deleted [45]. Other studies showed that the size of EEEV plaques in BHK-21 cells is significantly reduced after 10 serial passages in avian (PDE) or mosquito (C7-10) cells as compared to the parental strain. The strain resulting from alternating the cycle between avian and mosquito cells showed a less important reduction of plaque size [54] Virulence of VSV decreased after persistent infection in LL-5 and the size of plaques in BHK-21 cells decreased over the passages [53]. A study published in 1947 on BTV showed that virulence is attenuated after serial passages in embryonated eggs. The attenuated strain was injected into Merino sheep which showed no clinical signs and developed a solid immunity to a challenge with a virulent strain [55]. This study established the basis for developing live attenuated vaccines for sheep [56]. Further studies described methods of BTV attenuation, including serotypes 4, 9 and 16. Attenuation of these strains was accomplished by serial in vitro passages. BTV-4 and BTV-16 were passaged in Vero cells and BTV-9 in BHK-21 cells. During the process of attenuation, both replication capacity and pathogenicity were assessed in bovine foetal aorta endothelial cells (BFA) and new-born mice, respectively. The authors showed that attenuated BTV by in vitro serial passages had a reduced capacity to replicate in BFA cells and failed to kill new-born mice [57].

In DENV serially passaged in Huh-7 mammalian or C6/36 mosquito cells, more substitutions were identified in the genome of the serially passaged virus in mammalian (Huh-7) than that passaged in mosquito cells (C6/36) or by alternation [58]. During a DENV outbreak at the beginning of the 21st century, the virus was sequenced in naturally infected mosquitoes and patients. The sequence variation is lower in mosquitoes than in patients. A complementary study was performed in laboratory with experimentally infected mosquitoes by intrathoracic injection. Comparable results to those obtained with field-caught mosquitoes were observed. The authors concluded that mosquitoes play a role in stabilising the genome sequence and contribute to transmission of a DENV dominant variant [59]. Another study of CHIKV genetic variability was conducted. Serial and alternated passages of CHIKV in Aag2 and/or BHK-21 cells resulted in fewer amino acid substitutions in invertebrate cells than in mammalian cells or alternation of cells. Five substitutions were identified. Three substitutions were common to CHIKV propagated serially in mammalian cells or by alternation of cells. The genome of CHIKV which was propagated serially in invertebrate cells had only one substitution identified [60]. These results suggest that passages in invertebrates, constrain virus evolution significantly more than during host cycling. In a distinct study involving CHIKV, substitutions were found more frequently during serial mammalian passages than in alternated passages. Furthermore, mutations identified during serial passages were less viable than those found during alternation. All observed sequences in serial passages did not end up in viable virus particles. Genetic diversity is not synonymous with fitness. In alternating cycles, the virus has to maintain high replication competence and high fitness of variants [52]. Ciota et al. [61] assessed the role of the mutant spectrum in adaptation and replication of WNV. The virus was passaged 40 times in mosquito cells (C6/36) or 20 times in chicken embryo fibroblasts (DF-1). An alternated passage was performed in DF-1 following 39 passages in C6/36. Throughout the 40 passages in C6/36, genetic variability of WNV increased significantly but decreased drastically after one passage in DF-1 cells. Over the passages in DF-1 cells, the virus titer reached its peak earlier, as compared to the parental strain. In addition, the genotypic heterogeneity was slightly lower compared with mosquito-derived WNV. These results suggest that the size of the mutant spectrum depends on the host cell [61]. In the latter study, the results support the “trade-off” hypothesis. Vertebrates likely play the role of a genetic bottleneck for this virus, while invertebrate play an essential role in expanding genetic variability of arboviruses.

Genetic diversification of BTV-17 was assessed. Ten serial and alternated passages were performed in cells derived from *Culicoides sonorensis* (cell line CuVaW3) and/or bovine pulmonary artery endothelial cells (BPAEC). The consensus nucleotide sequences from serial or alternated passages was 100% identical to that of the parental strain in all segments, except segments 5 and 10 (>99.8%). After one passage in CuVaW3 or BPAEC cells, amino acid substitutions were found in segments 5 (I229R) and 10 (A360G). These changes were observed in the sequences of segments 5 and 10 of BTV-17 passaged serially or by alternation in the two cell types. Genetic diversity of BTV-17 remains stable independently of cell types or passages serially or by alternation in a mammalian and/or insect cell lines. The parental strain was isolated in BHK-21 cells from an infected sheep. The virus seemed to be subjected to a strong selection in BHK-21 cells and amino acid substitutions in segments 5 and 10 resulted from the adaptation of BTV-17 in CuVaW3 and BPAEC cells [62]. By contrast, another study, conducted with BTV-3, showed that genetic diversity was larger after one passage in *Culicoides sonorensis* (KC) cells when compared with the parental strain, which was isolated from sheep blood. In addition, after one passage in BSR (a clone of BHK-21) cells, diversity dropped in all segments. The passage history of the BTV-3 strain was distinct from that of BTV-17. The blood from BTV-3 infected sheep was inoculated into embryonated chicken eggs (ECE) or KC cells. After passage in ECE (BTV-3E1), the virus was further grown in C6/36 mosquito cells (BTV-3E1/C6-1) followed by three additional passages in BSR cells (BTV-3E1/C6-1/BSR1, BTV-3E1/C6-1/BSR2 and BTV-3E1/C6-1/BSR3). The BTV-3 grown in KC cells (BTV-3KC1) was further passaged three times in BSR cells (BTV-3KC1/BSR1, BTV-3KC1/BSR2 and BTV-3KC1/BSR3). Mutations were observed in VP2 (segment 2) and NS1 (segment 5) of BTV-3E1/C6-1/BSR2 or BTV-3E1/C6-1/BSR3. Other mutations were observed in VP5 (segment 6) and VP6 (segment 9) of BTV-3KC1/BSR2 or BTV-3KC1/BSR3. Non-synonymous nucleotide substitutions (amino acids changes: Q169R and M5I) were found in segment 8 (encoding NS2) of all BSR passages. The authors suggested that the various changes are likely involved in virus attenuation and/or host specialisation [63].

In summary, the trade-off hypothesis seems to be compatible with many of studied arboviruses except DENV and CHICKV. The evolution of these two viruses is constrained in their arthropod vector by contrast to WNV and the opposite is observed for WNV, for instance.

During in vitro experimental evolutionary studies of arboviruses, care should be taken when deciding upon which multiplicity of infection (MOI) to use. Indeed, when cells are infected at a high MOI, defective interfering particles (DI particles) are likely produced. DI particles were observed for several viruses belonging to different families such as the *Flaviviridae, Coronaviridae, Togaviridae, Paramyxoviridae, Rhabdoviridae* and *Reoviridae* [64]. Undiluted passages of viruses lead to accumulation of DI particles and result in a decrease of viral fitness. The DI particles can interfere with the standard virus replication, which can lead to rapid genome evolution. Studies of EEEV and VSV have shown that DI particles were preferentially replicated [46,65].

#### 2.2.2. In Vivo Studies

Conducting in vivo studies with arboviruses to assess their genetic variability is a less frequent setting, due to the complex nature of these studies. In fact, these studies are subject to approval by ethics committees, require high containment animal facilities, rely on the availability of a relevant animal model and are time consuming, to cite a few essential conditions. Usually, the design of in vivo experiments is similar of those done in vitro: ten serial or alternated passages are made in mice or hamster and arthropods. For specific viruses, the results obtained in vitro are different from those passaged in vivo, yet certain characteristics are common to both situations. Host specialisation in serial passages was observed and assessed by measuring replication kinetics and viraemia in vertebrates. A study by Coffey et al. (2008) [48], focussed on VEEV, showed host specialisation of two strains (ID and IC) in mice or hamsters and mosquitoes. The viral strains derived from 10 serial passages in mosquitoes induced low levels of viraemia in mice and resulted in higher infection rates in mosquitoes as compared to the first passage. After 10 sequential passages, viraemia in vertebrates peaked earlier. Characteristics of viraemia were similar to those of parental strain in viruses alternated between mice or hamsters and mosquitoes [48]. These results support that alternation of host likely constrains evolution of a large number of arboviruses. Moreover, a non-synonymous nucleotide substitution was observed after 10 serial passages in mice in the sequence encoding the nsP4 polymerase. The frequency of mutated nsP4 position 6174 (R6174S) increased over the passages and seemed to correlate with earlier and higher levels of viraemia in mice. In addition, during serial passages in hamsters, a distinct amino acid substitution was observed in nsP4 sequence. Serial VEEV passaged in mosquitoes resulted in a synonymous substitution at nucleotide position 123 of the nsP1 sequence. The viral population with the latter substitution occurred at very low frequency, and it was only until the eight passage that its frequency increased, becoming relatively frequent at the tenth passage [48].

The viruses resulting from in vitro passages are usually further tested in mice and arthropods to evaluate their phenotypes. Serial in vitro passages result in a decreased fitness (and consequently resulted in lower levels of viraemia) in vivo as compared to the parental strain. It has been shown for RRV after serial passages in BHK-21 cells, where virulence in mice decreased. By contrast, animals died earlier when RRV was serially passaged in mice, thus selecting more virulent strains [66]. In a study by Moutailler et al. (2011) [45] focussing on RVFV, the strains and clones resulting from serial passages in BHK-21 and Aag2 cells did not cause death in mice. The parental strain and those resulting from alternated passages induced 100% mortality in less than nine days and behaved similarly in animals. Mice were also infected with viruses derived from serial passages in BHK-21 and Aag2 (deleted *NSs*). These mice were further challenged with the virulent parental strain and all mice were protected from the lethal infection [45].

Subpopulations generated upon infection of mammals/arthropods likely influence the outcome of infection and onward transmission. Despite this knowledge, virus quasi-species were not analysed in a mammalian host/arthropod vector in the majority of published studies. The extent of genetic variation is virus dependent. For instance, analysis of blood collected from mice inoculated with Liao ning virus (LNV) or Banna virus (BAV) (two members of genus *Seadornavirus*) showed significantly higher sequence variations for LNV than BAV [67].

### 2.3. Genetic Variability of Tick-Borne Viruses

Studying the genetic variability of arboviruses mainly focussed on mosquito- or sandfly-borne viruses, especially those belonging to genera *Flavivirus* and *Alphavirus.* There is a lack of data for tick-borne viruses, yet the “trade-off” hypothesis was extrapolated to all arboviruses, including tick-borne viruses. While mosquitoes, sandflies and ticks do not have the same lifecycle and lifestyle, tick-borne viruses are longer associated with their vectors than mosquito- or sandfly-borne viruses, due to vector life span.

Limited studies were conducted on tick-borne viruses such as TBEV and CCHFV. In 2007, Romanova et al. [68] conducted a study on TBEV in order to compare a tick-adapted strain, variant M to a parental strain EK-328. Variant M is the result of 17 serial passages in *Hyalomma marginatum marginatum* and five passages in mouse brains. The first observed difference between the two strains was the size of plaques: EK-328 produced 7-mm plaques, whereas variant M made 1-mm plaques in majority and few 7-mm plaques (ratio 100:2). The tick-adapted strain was less virulent in mice than the parental strain and viral titers in ticks were higher for variant M after seven days post-inoculation. Furthermore, growth kinetics in PEK cells (pig embryo kidney cells) differed. The yield of variant M in cell culture supernatants was 100 times lower than from EK-328 at 22 h post-infection and did not increase at 60 h post-infection. A full genome sequencing identified fifteen nucleotide substitutions in variant M, six of which were non-synonymous (resulting in changes in protein sequences for E, prM, NS2A and NS4A). Changes in the E protein sequence correlated with the smaller plaque size. Depending on the positions impacted by amino acid changes, they were also linked to stronger or lower affinities of the virus for cellular heparan sulphate and the binding of virions to cellular glycosoaminoglycans. Moreover, the neuroinvasive properties in mice have been impacted for specific substitutions in E protein [68]. A Further study was conducted with Langat virus (LGVT, a tick-borne flavivirus), during which two variants were generated. After 20 serial passages in mouse neuroblastoma (MNB) or *Ixodes scapularis* tick (ISE6) cells, variants MNBp20 and ISEp20 were, respectively, identified. Host specialisation was observed in both cell types. The full-length genome sequences of the two variants were determined in order to identify genetic changes associated with adaptation. Amino acid changes found in both viruses were identified in structural and non-structural proteins. The mouse cell adapted virus had amino acid changes in the E (E277K and Y438Y/H), NS4A (E33G) and NS4B (K164K/R) proteins. The tick cell adapted virus had substitutions in prM (K115E), NS3 (F604F/L) and NS4A (A81A/V) proteins. It was concluded that all five proteins E, prM, NS3, NS4A and NS4B are involved in host adaptation of LGTV [69].

In 2016, Han Xia et al. [31] studied how a tick vector and an animal host shape the genome plasticity of CCHFV. They showed transmission of the virus to *Hyalomma marginatum* ticks upon feeding on experimentally infected mice, as well as transstadial transmission in ticks. They also conducted a study to assess the impact of long-term association of CCHFV with ticks on the viral genome. For that purpose, they fed nymphs on infected mice and conserved them one year after they moulted into the adult stage, before RNA extraction from ticks and mice, followed by NGS sequencing. Substitutions in CCHFV genome were observed only in tick samples. In total, fourteen mutations were identified: one, four and nine in the S, M and L segments, respectively, as compared to the virus from infected mice. Specific substitutions were common in the two groups of ticks which were used in the experiment, while other substitutions in segment L were found in one-year-old ticks only. Among the fourteen substitutions, four were non-synonymous in segments M and L. In addition, CCHFV genome diversity was higher in ticks than in mice [31]. These results showed that tick vectors play a principal role in expanding the genetic variability of tick-borne viruses. To assess the “trade-off” hypothesis for CCHFV, it would have been valuable to infect mice using the adult ticks which moulted from infected nymphs. The sequence of the resulting virus in mice would have been a source of valuable information, to prove or disprove the “trade-off” hypothesis for this tick-borne virus.

## 3. Arbovirus–Vector as a Complex System

### 3.1. Co-Evolution between Arboviruses and Arthropod Vectors

There are different views as to whether arboviruses co-evolved or not with their arthropod vectors. Looking at genomes of arboviruses in genera where tick-borne, mosquito-borne, sandfly-borne and/or *Culicoides*-borne are encountered such as the flaviviruses or the orbiviruses, different observations are made. One of the important observations pertains to the G + C content of their genomes. For flaviviruses, the G + C content of insect-only viruses is in the range of 50–53%, that of mosquito-borne viruses is in the range of 47–53%, for tick-borne viruses it is 53–54% and for no known vector viruses (NKV) it is 43–48%, with the exception of Tamana bat virus (TABV) genome where G + C content is ~38%. As for orbiviruses, the G + C content of mosquito-borne orbiviruses ranged from 35% to 41%. The G + C content of *Culicoides*-borne viruses ranged from 40% to 45%. For tick-borne orbiviruses, the G + C content ranged from 52% to 58%. The G + C content of arthropod vector genomes is ~56% for ticks, ~39–42% for *Culicoides* and 35–38% for mosquitoes [70,71]. There are clear differences between the two genera. Regarding flaviviruses, only tick-borne viruses have a G + C content which is similar to that of their tick vectors (~57%). G + C content of insect only flaviviruses is very similar to that of tick-borne viruses and for mosquito-borne flaviviruses the G + C content is largely overlapping with that of the insect-only and tick-borne flaviviruses. This largely contrasts with orbiviruses where G + C content is very similar to that of their respective arthropod vectors. These observations and clear specialisation, among other findings, provided ground to suggest that orbivirus ancestors were arthropod viruses which coevolved with their vectors and adapted to vertebrate hosts [70].

Earlier phylogenetic studies classified flaviviruses within four groups: two mosquito-borne groups, a tick-borne group and a group where viruses are designated as non-vectored or having no known vectors (NKV) [72]. Phylogenetic studies based on NS5 gene showed it to contain insufficient signal to corroborate a specific tree topology. Phylogenetic trees based on the NS3 protein or full polyprotein sequences do suggest that mosquito-borne flaviviruses root all other flaviviruses. They also suggest that these viruses adapted later on to tick transmission [73]. According to the gene used for building a phylogenetic tree for flaviviruses, the phylogenetic relationships between the various groups differ. Trees based on NS5 protein sequence showed that mosquito-borne viruses and tick-borne viruses are sister groups whereas trees based on NS3 protein sequence showed that tick-borne viruses and NKV are sister groups, with insect-specific flaviviruses as outgroup for both [74]. Other studies suggest that NKV flaviviruses do not form a single phylogenetic group. They suggest that possible recombination events may explain discrepancies of clustering in phylogenetic trees (based on full-length polyproteins or specific genes such as the envelope, NS3 or NS5). These studies suggested that *Yokose* (YOK), *Sokoluk* (SOK) and *Entebbe bat* (ENT) *virus* seem to cluster with mosquito-borne flaviviruses. The other NKV viruses either cluster with tick-borne flaviviruses (Envelope, NS3 or polyprotein trees) or are basal to mosquito and tick-borne flaviviruses (NS5 trees) [75]. It was also suggested that YOKV, SOKV and ENTV may be arboviruses instead of being NKVs and that based on usage of dinucleotide frequencies they do not seem to be specifically adapted to vertebrate hosts. Only TABV seems to be a vertebrate specific virus, having a significantly lower CpG dinucleotide frequency than insect-specific flaviviruses or arthropod/vertebrate flaviviruses [75].

Whether flaviviruses co-evolved with their arthropod vectors (i.e., originated before the split of invertebrates and vertebrates during evolution) or they have originated in one group or the other followed by co-adaptation, remains a debatable question. However, the high G + C content of most flavivirus genomes (including insect-only viruses) raises questions as to whether flaviviruses did originate in ticks or if this trait provided advantage to the dispersal of flaviviruses among various arthropod vectors. Certain insect-borne viruses are capable of infecting and replicating in tick cell lines. Semliki Forest virus (SFV) and Venezuelan equine encephalitis virus (VEEV), two mosquito-borne alphaviruses, replicate well in tick cell lines *Rhipicephalus decoloratus* BDE/CTVM16 and *Rhipicephalus appendiculatus* RAE/CTVM1 [76,77]. The G+C content of SFV genome is ~53% and that of VEEV is ~50%. Replication of 13 flaviviruses including DENV, WNV, SLEV, yellow fever virus (YFV), TBEV, POWV, Louping ill virus (LIV), Negishi virus (NGV) and LGTV was compared in mosquito and tick cell lines. The mosquito-borne viruses replicate efficiently in mosquito cells and it was shown that WNV is capable of infecting and replicating in four tick cell lines. The tick-borne viruses replicate only in tick cell lines, except LGTV where signs of infection were observed in C6/36 [77]. A previous study using Singh’s non-cloned *Ae. albopictus* cells (from which C6/36 were derived) failed to show replication of LGTV [78]. Two NKV flaviviruses were also tested in this study and none replicated in any cell line [77]. The G + C contents of DENV, SLEV and YFV are between 45% and 50%, whereas those of WNV, TBEV, POWV and LIV are up to 51%. Larvae and nymphs, but not adults, of *Amblyomma* species were shown experimentally to be susceptible for infection by WNV or SLEV [79]. These results highlight important differences between flaviviruses in terms of their capacity to potentially infect a distinct arthropod vector.

*Colorado tick fever virus* (CTFV) and *Eyach virus* (EYAV) are two species belonging to genus *Coltivirus* (family *Reoviridae*). Replication of CTFV or EYAV in mosquito and/or tick cell lines has been explored. CTFV was grown in *Ae. albopictus* cells over seven weeks and the titer reached ~10^6^ pfu/mL after six weeks, declining afterwards [80]. As compared to titers of the same virus grown mammalian cells (~10^8^ pfu/mL), the difference represents approximately 99% less virus in mosquito cells. Limited replication of a EYAV-Fr578 was shown in C6/36 cells [81]. A strain of Kemerovo virus (KEMV, genus *Orbivirus*) was grown both in *Ae*. *albopictus* C6/36 cells and *I. ricinus* tick cells. While titers of approximately 10^7^–10^8^ pfu/mL were observed in tick cells, titers of 10^4^ or less were obtained with C6/36 (unpublished data), that is to say 99.9–99.99% less virus in mosquito cells.

In summary, the lower levels of replication of a given animal arbovirus in a heterologous cell culture system suggest that a heterologous arthropod would not act as a likely vector for that particular virus.

### 3.2. Role of Arthropods in Natural Survival and Spread of Arboviruses

Since the 1930s, several basic concepts were established for arbovirus–vector systems. Plant arbovirus associations with their competent insect vectors fall into circulative propagative, circulative non-propagative or non-circulative cycles. There are no examples of circulative non-propagative transmission of an animal arbovirus. Competent arthropod vectors of animal arboviruses ingest their blood meal and the virus replicates in arthropod cells and disseminate, reaching salivary glands to ensure transmission during the next blood meal. In addition, viruses must survive arthropods’ behavioural characteristics. Indeed, according to the arthropods, the conditions are not the same if it is an insect or an acarian (e.g., ticks). Contrarily to mosquitoes that can take several blood meals at the same life stage (adult female), hard ticks feed only once before moulting, meaning that viruses must survive upon moulting into the next stage. Moreover, several months can pass between two stages [82]. The extrinsic incubation period (EIP) corresponds to the time between the ingestion of pathogens and the transmission to a next host [83]. Viruses also have to establish strategies to survive in arthropod. Tick-borne viruses need to persist in ticks, for instance by infecting tissues which do not undergo histolysis during moulting [84]. Some tick-borne viruses can also be transmitted trans-ovarially and/or trans-sexually to the next generation. Vertical transmission and transmission by co-feeding between ticks contribute to the natural survival and spread of tick-borne viruses [82].

The EIP spans the time starting with initial virus attachment to midgut cells, replication and dissemination into the arthropod. These are key phases for virus transmission, requesting the crossing of biological barriers including the gut barrier until reaching salivary glands [83]. The gut barrier, a determinant of vector competence, controls the virus both qualitatively and quantitatively and defines the permissivity of an arthropod to a given virus. In addition, the process of blood digestion is important for a successful infection of the arthropod. Ticks are heterophagous, meaning that the first step of digestion is intracellular (midgut cells) while insects such as mosquitoes digest their blood meal in the lumen of midgut [84]. Tick gut cells were shown to be a site where Thogoto virus (family *Orthomyxoviridae*, genus *Thogotovirus*) genome segments reassort [85]. After crossing the gut, the virus replicates within. It is then released into the haemolymph circulation and is carried until the salivary glands. A new phase of replication takes place within these glands, provided that virus titers are sufficient enough in the haemolymph. A successful dissemination of the virus to key tissues of the arthropod is a parameter for studying vector competence. That was observed in CHIKV upon assessing virus fitness and vector competence of *Aedes* mosquitoes [60]. Furthermore, the concentration of ingested virus as well as ecological factors must be taken into consideration [83]. Antiviral mechanisms developed by hosts and vectors could be the source of genetic variability of viruses. Arthropod vectors developed various innate immune responses to limit or control viral infections. These responses include RNA interference (RNAi, generating short interfering RNA or siRNA), Jak-STAT (Vago), Nf-κB, Imd and Toll and autophagy. siRNA seems to be the most robust pathway to control virus after an infection, by degrading RNA and limiting replication [86].

One of the entomological variables which are considered for pathogen transmission is the vectorial capacity. This variable includes biotic (vectorial competence) and abiotic (geographical area, climate, population density, etc.) factors. Geographical distribution of vectors is influenced by climate change and human activities (farming, deforestation, urbanisation, international travel and trade, etc.). These parameters impact vector ecology and consequently human exposure to infection [87]. Abundance of vectors considerably impacts the emergence of a pathogen. This was observed during the CHIKV outbreak in the Indian Ocean in 2004, where the conditions were optimal (high abundance of *Ae. albopictus*) for the virus to emerge since the amino acid substitution A226V in the E1 envelope glycoprotein helped the virus to switch vector. *Aedes albopictus* has a larger geographical distribution than *Aedes aegypti*, which is almost absent in the Indian Ocean islands [88,89]. The abundance of vectors must be considered, taking into account other abiotic factors such as climate, temperature and seasonality. The emergence of specific viruses can be facilitated by optimal climate conditions and the presence of local competent vectors. The emergence of BTV in the north of Europe in summer of 2006 is a typical example, where transmission occurred by the local midge population. Other examples include ZIKV in Brazil in 2015 and DENV in India in 2015 [90]. Arboviruses and their vectors must be considered as complex systems where several parameters affect their dynamics. The genetic variability of arboviruses strongly influences this system. The genetic variability of insect-borne arboviruses has been assessed for a range of viruses. Additional focus should be placed on tick-borne viruses to bring these studies up to speed.

## 4. Conclusions

Nowadays, vector-borne diseases are important both medically and economically. Characteristics and molecular aspects of arboviruses were identified during multiple studies aiming to assess their genetic variability and in particular the insect-borne viruses (Table 3). Viruses host-specialise in serial passages or are able to infect two distinct hosts or cell types in alternating passages. The loss or gain of fitness can be evaluated by studying different in vitro and in vivo characteristics: (i) growth kinetics (time course study of virus titers); (ii) capacity to infect distinct cell types; (iii) induction of viraemia in experimentally infected mice (strain dependent: peaking earlier or later, being lower or higher as compared to the parental strain); (iv) infection rates of arthropods; and (v) virus dissemination in arthropods (see references in Table 3). The experimental set-up influences virulence. Hence, virulence tends to decrease in vitro set-ups, whereas the contrary is observed in vivo set-ups. Sequence analyses, in particular deep sequencing, helped linking changes in virulence or fitness to specific viral genes. Virulence can be measured by evaluating morbidity/mortality rates in vertebrates. Genes and encoded proteins that are responsible for interactions between viruses and cells, particularly envelope or outer capsid proteins, accumulate mutations which allow the virus to infect efficiently one cell type or distinct vector species.

Several studies of the genetic variability have been conducted with mosquito- or sandfly-borne viruses in order to understand their capacity to emerge and potentially adapt to new hosts, notably by generating virus quasi-species. The latter are produced and compartmentalised into different tissues/organs of the infected subject. Quasi-species should be taken into account upon studies focussing on characterisation of genetic diversity, as important drivers predicting virus emergence (see references Table 3). Alternating host transmission cycle may constrain arbovirus evolution as suggested by the “trade-off” hypothesis. However, this hypothesis was extrapolated to all animal arboviruses including tick-borne viruses where there is a paucity of information regarding genetic variability. Insects and ticks do not have the same life cycle and behavioural characteristics and viruses might not be exposed to the same pressure depending on the arthropods they infect. In total, 170 tick-borne viruses were identified so far and they are understudied compared to insect borne viruses. During the last few decades, we faced tick-borne virus emergence and it is necessary to understand and anticipate such an emergence. Strategies have been developed to study mosquito- or sandfly-borne viruses. These strategies are useful to build on and study the genetic variability of tick-borne viruses. They will be useful to attempt identifying genes or genetic elements, which are essential for replication in both vectors and hosts and which potentially drive virus emergence into new geographic areas.

## Figures and Tables

**Figure 1 pathogens-09-00915-f001:**
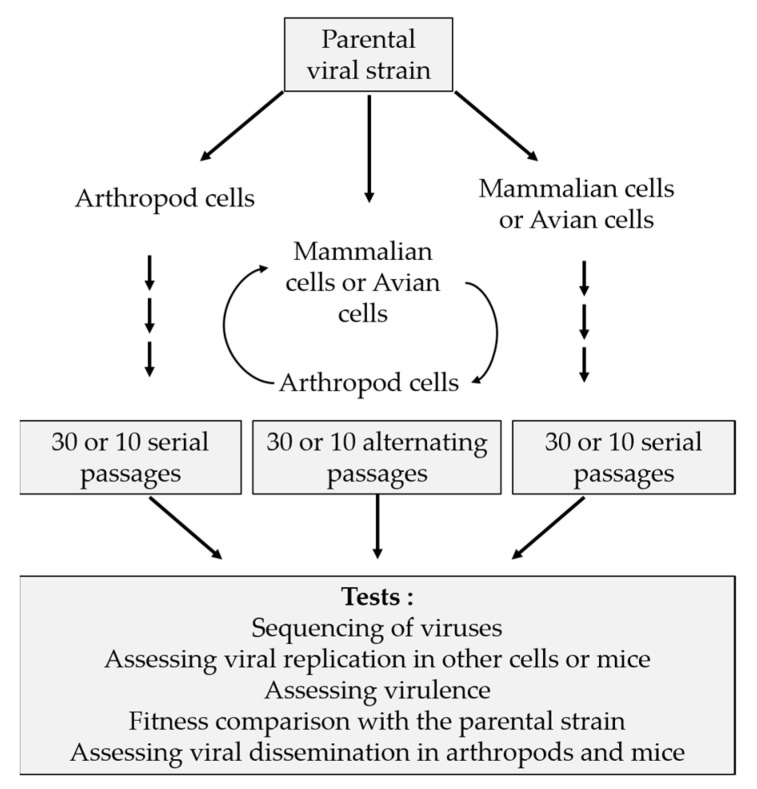
Experimental strategy to test the genetic variability of arbovirus.

**Table 1 pathogens-09-00915-t001:** Families and genera of tick-borne viruses.

Order	Family	Genus	Genome
*Asfuvirales*	*Asfarviridae*	*Asfivirus*	dsDNA
*Articulavirales*	*Orthomyxoviridae*	*Thogotovirus*	6 ssRNA genome segments 8 ssRNA genome segments
		*Quaranjavirus*	
*Bunyavirales*	*Nairoviridae*	*Orthonairovirus*	3 ssRNA genome segments
	*Peribunyaviridae*	*Orthobunyavirus*	
	*Phenuiviridae*	*Phlebovirus*	
*Mononegavirales*	*Nyamiviridae*	*Nyavirus*	Non-segmented linear ssRNA
	*Rhabdoviridae*	*Ledantevirus*	
		*Vesiculovirus*	
*Amarillovirales*	*Flaviviridae*	*Flavivirus*	Linear ssRNA
*Reovirales*	*Reoviridae*	*Orbivirus*	10 dsRNA genome segments
		*Coltivirus*	12 dsRNA genome segments

ss, single-stranded; ds, double-stranded.

**Table 2 pathogens-09-00915-t002:** Examples of tick-borne viruses.

Virus	Family	Genus	Vector	Vertebrate Reservoir	Geographical Distribution	Disease	References
*Tick-borne encephalitis virus* (TBEV)	*Flaviviridae*	*Flavivirus*	*I. ricinus, I. persulcatus*	Bank vole	Europe Asia	Fever to encephalitis	[14,15,16]
*Powassan virus* (POWV)			*I. scapularis*, *I. cookie*	Rodents	North America	Neurological disorders	[17,18]
*Omsk haemorrhagic fever virus* (OHFV)			*D. reticulatus, D. marginatus* and *I. persulcatus*	Rodents	Western Siberia in Russia	Haemorrhagic fever	[19]
Alongshan virus (ALSV)	Unclassified flavi-like virus		*I. ricinus, I. persulcatus*	?	China Finland	Fever	[20,21,22]
*Colorado tick fever virus* (CTFV)	*Reoviridae*	*Coltivirus*	*Dermacentor, Ixodes, Haemaphysalis* and *Otobius*	Rodents and deer species	North America	Fever to encephalitis	[23,24]
*Bourbon virus* (BRBV)	*Orthomyxoviridae*	*Thogotovirus*	*Amblyomma americanum*	?	North America	Nausea, weakness, pains and leukopenia, lymphopenia, thrombocytopenia, hyponatremia	[25,26]
*Heartland virus* (HRTV)	*Phenuiviridae*	*Phlebovirus*	*Amblyomma americanum*	?	North America	Fever, fatigue, anorexia and thrombocytopenia	[27]
*Severe fever with thrombocytopenia syndrome virus* (SFTSV)			*Haemaphysalis longicornis* and *Boophilus microplus*	?	Asia	Fever, fatigue, anorexia and thrombocytopenia	[27]
*Nairobi sheep disease virus* (NSDV)	*Nairoviridae*	*Orthonairovirus*	*Rhipicephalus appendiculatus, Haemaphysalis intermedia*	Sheep and goats	Africa, Asia	Fever and haemorrhagic gastroenteritis, abortion, and high mortality	[28,29]
*Crimean-Congo haemorrhagic fever virus* (CCHFV)			*Hyalomma spp.*	Cattle, goats, sheep and hares?	Africa, Southern and Eastern Europe and Asia	Haemorrhagic fever	[30,31,32]
*African swine fever virus* (ASFV)	*Asfarviridae*	*Asfivirus*	*Ornithodoros spp.*	Swine	Europe Asia	In animal: fever, depression, anorexia, abortion in gestating female	[33]

**Table 3 pathogens-09-00915-t003:** Summary of all studies described in this review.

Virus		In Vitro, In Vitro/In Vivo, In Vivo	Findings	References
WNV	*Flavivirus*	in vitro	Specialisation to a single cell type in serially passaged virus Increased genetic variability after serial passages in mosquito cells as compared to mammalian-derived strain	[49,61]
SLEV		in vitro	Specialisation to a single cell type in serially passaged virus	[49]
RBGV		in vitro	Adaptation to cell culture in mammalian cells at high temperature Role of NS3 in host range	[51]
DENV		in vitro	Lower number of substitutions in mosquito-cell serially passaged virus as compared to alternated virus	[58,59]
ZIKV		in vitro	Substitutions in envelop protein that give benefits for replication in a cell type	[50]
TBEV		in vivo	Tick-adapted strain less virulent in mice than the parental strain Mutations in E, prM, NS2A ans NS4A—a role of E protein in fitness	[68]
LGVT		In vitro	Specialisation to a single cell type in serially passaged virus Mutations in E, prM, NS3, NS4A and NS4B—a role of these proteins in host adaptation	[69]
EEEV	*Alphavirus*	in vitro	Specialisation to a single cell type in serially passaged virusSame characteristics as parental strain in alternated passages	[46,54]
VEEV		in vitro	Specialisation to a single cell type in serially passaged virus—Increased binding efficiency to mammalian cells	[48]
		in vivo	Host specialisation following serial passages in mammals	
SINV		in vitro	Faster growth kinetics in single cell type	[47]
CHIKV		in vitro	Specialisation to a single cell type of serially passaged virus Same characteristics as parental strain in alternated passages Fewer amino acid substitutions in invertebrate cells than in mammalian cells and alternation of cells	[52,60]
RRV		in vitro	Decreased virulence following serial passages	[66]
		in vivo	Increased virulence following serial passages	
RVFV	*Phlebovirus*	in vitro	Specialisation to a single cell type and decreased virulence following serial passages Faster growth kinetics in the same cell type	[45]
		in vitro/in vivo	Mammalian-cell adapted strain less virulent in mice than parental and alternated strains	
VSV	*Rhabdovirus*	in vitro	Decreased virulence following serial passages Decreased fitness in serial passages in arthropod cells and alternating passages Increased fitness following serial passages in mammalian cells	[53]
BTV	*Orbivirus*	in vitro	Decreased virulence in serially passaged virus Increased genetic diversity in arthropod cells	[55,57,63]
CCHFV	*Orthonairovirus*	in vivo	Substitutions in viral genome observed only in tick samples Genetic variability higher in ticks than in mice	[31]

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
