# Peer review of "Strategies for Assessing Arbovirus Genetic Variability in Vectors and/or Mammals"

_pathogens, 2020, doi:10.3390/pathogens9110915_

Round 1
Reviewer 1 Report
This is a very interesting review manuscript which describes the state of the art regarding the strategies developed to study genetic variability of arboviroses in vivo and in vitro, and their potential applications to tick-borne viruses less studied than mosquito-borne viruses. Specificities of tick-borne viruses’ lifecycle and lifestyle are highlighted as well as arboviruses and vectors as complex and diverse system.
This manuscript provides a nice overview of this subject. I have only a few, relatively minor, requests for changes.
Specific comments:
- Although others manuscripts by these same authors are referenced, I think that a previous review overlapping this subject by Forrester et al. (Forrester, N.L.; Coffey, L.L.; Weaver, S.C. Arboviral Bottlenecks and Challenges to Maintaining Diversity and Fitness during Mosquito Transmission. Viruses 2014, 6, 3991-4004.) that focus mainly on WNV, VEEV and CHIKV, would be important to include in the overview of the presented data/references.
- Table 2 – in Disease column please use capital letter in the start of clinical description for each virus, in this way, the same rule will be address to all table lines and the reading becomes easier.
-Table 3 – please remove the line in References column when in vitro and in vivo studies have the same reference (for each virus: VEEV, RRV and RVFV). Please remove the line in Virus column, RRV (separating in vitro/in vivo) studies.
-Line 411 – please erase “final dot” after 2004
Author Response
We would like to take the opportunity to thank both reviewers for their support and very useful comments, which helped improving the manuscript.
This is a very interesting review manuscript which describes the state of the art regarding the strategies developed to study genetic variability of arboviroses in vivo and in vitro, and their potential applications to tick-borne viruses less studied than mosquito-borne viruses. Specificities of tick-borne viruses’ lifecycle and lifestyle are highlighted as well as arboviruses and vectors as complex and diverse system.
This manuscript provides a nice overview of this subject. I have only a few, relatively minor, requests for changes.
Specific comments:
- Although others manuscripts by these same authors are referenced, I think that a previous review overlapping this subject by Forrester et al. (Forrester, N.L.; Coffey, L.L.; Weaver, S.C. Arboviral Bottlenecks and Challenges to Maintaining Diversity and Fitness during Mosquito Transmission. Viruses 2014, 6, 3991-4004.) that focus mainly on WNV, VEEV and CHIKV, would be important to include in the overview of the presented data/references. Thank you for this reference, it was added at lines 115-116 as recommended
- Table 2 – in Disease column please use capital letter in the start of clinical description for each virus, in this way, the same rule will be address to all table lines and the reading becomes easier. The modification in the table was made as requested
-Table 3 – please remove the line in References column when in vitro and in vivo studies have the same reference (for each virus: VEEV, RRV and RVFV). Please remove the line in Virus column, RRV (separating in vitro/in vivo) studies. Correction has been made as requested
-Line 411 – please erase “final dot” after 2004 Correction has been made
Reviewer 2 Report
The aim of the manuscript of Migné et al is the description of the strategies developed to study the genome evolution of insect-borne viruses, their potential application to tick-borne viruses, and the complexity that characterized different arboviruses-vector systems. To this end, after a general introduction of vector-borne diseases, Authors described the problems related to virus transmitted by ticks highlighting their impact on the global health and economy. Furthermore, Authors provide an overview on ticks and tick-borne viruses before to present strategies and results obtained in the field of genome evolution of insect-borne viruses. Finally, Authors reported data indicating the complexity of the vector-virus system.
In general, the review is well written and could be improved with a paragraph (such as perspective or future directions) suggesting the best approach to study viral evolution and adaptation in ticks to show how the experience acquired on insect-borne viruses can be translated to tick-borne viruses.
Furthermore, additional comments are following reported:
Major
In Tab.1 the column “Family” included also some “Orders” making the interpretation not obvious. I suggest to add one column reporting the Order associated to each viral family indicated in the table.
In Tab. 2 the name of the column “Reservoir” is not appropriated. Reservoir indicates the animal host allowing the persistence of a pathogen in the environment. For some viruses, such as TBEV, it has been reported that some ticks are the reservoir while vertebrate are only hosts for viral amplification and diffusion. So, the name of column should be changed in “Host” and one more column should be added indicating the viral reservoir. In the case of ticks, not all the vectors can be also reservoirs.
In addition, the list of viruses could be ordered for family.
Line 76: “for outbreaks”. Please indicate humans/animal/both for clarity.
Lines 107-109: starting from “In addition, ….” Till reference 37. Rewrite for clarity
Line 167: clarify if A5716G is a mutation selected during the in vitro evolution
Lines 212-214: Authors reported that the study on WNV supported the trade-off hypothesis with arthropods involved in the increasing of genetic variability while vertebrates reduce act as bottleneck. However, all the studies reported in lines 189-205 on DENV, VSV, and CHICK supported the opposite situation. Authors should comment better this aspect describing more in details the experimental setting or the reasons for this difference.
Lines 215-221: This section is not clear. Rewrite for clarity adding more details to understand the relevance of DI particles on the evolutionary studies.
Line 222: the “In vivo studies” paragraph should be rewrite; it is confusing. Authors should separate studies performed only in vivo (evolution step and biological effects) from studies in which the evolution stage was performed in vitro and the biological effects were evaluated in vivo. This can increased the clarity.
Furthermore, more details should be added on the “only in vivo” studies highlighting the differences with the in vitro or in vitro/in vivo studies reported in literature. Line 255: the number of the paragraph 2.2.3 “genetic variability of tick-borne viruses” should be changed in 2.3. It is independent by the paragraph 2.2 “Genetic variability studies of insect-borne viruses” focalized on insects and not ticks.
Line 366: a final comment about the different ability of vector-borne viruses to replicate in insect and/or tick cell lines can help the reader to understand the message that Authors want to provide.
---------------------
Minor
Line 134: change “RVFV” as “(RVFV)” as for the others viruses.
Line 223: change “In vivo” with “in vivo”
Author Response
We would like to take the opportunity to thank both reviewers for their support and very useful comments, which helped improving the manuscript.
The aim of the manuscript of Migné et al is the description of the strategies developed to study the genome evolution of insect-borne viruses, their potential application to tick-borne viruses, and the complexity that characterized different arboviruses-vector systems. To this end, after a general introduction of vector-borne diseases, Authors described the problems related to virus transmitted by ticks highlighting their impact on the global health and economy. Furthermore, Authors provide an overview on ticks and tick-borne viruses before to present strategies and results obtained in the field of genome evolution of insect-borne viruses. Finally, Authors reported data indicating the complexity of the vector-virus system.
In general, the review is well written and could be improved with a paragraph (such as perspective or future directions) suggesting the best approach to study viral evolution and adaptation in ticks to show how the experience acquired on insect-borne viruses can be translated to tick-borne viruses.
Furthermore, additional comments are following reported:
Major
In Tab.1 the column “Family” included also some “Orders” making the interpretation not obvious. I suggest to add one column reporting the Order associated to each viral family indicated in the table. Correction has been made as requested
In Tab. 2 the name of the column “Reservoir” is not appropriated. Reservoir indicates the animal host allowing the persistence of a pathogen in the environment. For some viruses, such as TBEV, it has been reported that some ticks are the reservoir while vertebrate are only hosts for viral amplification and diffusion. So, the name of column should be changed in “Host” and one more column should be added indicating the viral reservoir. In the case of ticks, not all the vectors can be also reservoirs.
In addition, the list of viruses could be ordered for family. We meant to describe the vertebrate reservoir. This is now precised in the table. We agree with the comment of the reviewer that ticks can play a role of reservoir. This is why we have now added a sentence at lines 70-71 about ticks playing the role of reservoir for tick-borne viruses. We would like to avoid listing hosts because there are numerous “accidental hosts”, thus crowding the table. For instance TBEV can infect humans, horses, goats, cows,… It’s not the topic of this review to describe hosts in an exhaustive manner.
Line 76: “for outbreaks”. Please indicate humans/animal/both for clarity. Correction has been made, as requested
Lines 107-109: starting from “In addition, ….” Till reference 37. Rewrite for clarity This passage was rewritten for clarity as requested by the reviewer
Line 167: clarify if A5716G is a mutation selected during the in vitro evolution Correction has been made
Lines 212-214: Authors reported that the study on WNV supported the trade-off hypothesis with arthropods involved in the increasing of genetic variability while vertebrates reduce act as bottleneck. However, all the studies reported in lines 189-205 on DENV, VSV, and CHICK supported the opposite situation. Authors should comment better this aspect describing more in details the experimental setting or the reasons for this difference. We added comments and made further precisions, added information about more viruses in particular dsRNA virus such as bluetongue.
Lines 215-221: This section is not clear. Rewrite for clarity adding more details to understand the relevance of DI particles on the evolutionary studies. This passage was rewritten for clarity as requested.
Line 222: the “In vivo studies” paragraph should be rewrite; it is confusing. Authors should separate studies performed only in vivo (evolution step and biological effects) from studies in which the evolution stage was performed in vitro and the biological effects were evaluated in vivo. This can increased the clarity. Correction has been made. We have separated both parts.
Furthermore, more details should be added on the “only in vivo” studies highlighting the differences with the in vitro or in vitro/in vivo studies reported in literature. Line 255: the number of the paragraph 2.2.3 “genetic variability of tick-borne viruses” should be changed in 2.3. It is independent by the paragraph 2.2 “Genetic variability studies of insect-borne viruses” focalized on insects and not ticks. Correction has been made as requested. We added a supplementary study conducted with LGTV.
Line 366: a final comment about the different ability of vector-borne viruses to replicate in insect and/or tick cell lines can help the reader to understand the message that Authors want to provide. We added a final comment at lines 430-432.
---------------------
Minor
Line 134: change “RVFV” as “(RVFV)” as for the others viruses. Correction has been made
Line 223: change “In vivo” with “in vivo” Correction has been made
Round 2
Reviewer 2 Report
The new version of the manuscript includes all the suggested corrections. The manuscript is clear and and well written and, in my opinion, suitable for publication as it is stand.